# Correlation between the Concentration of Secondary Metabolites and Soil Microorganisms in *Sophora Koreensis* Nakai from Different Habitat

Hwa Lee and Eun Ju Cheong *

College of Forest and Environmental Science, Kangwon National University, Chuncheon 24341, Korea; dlghk018@kangwon.ac.kr
* Correspondence: ejcheong@kangwon.ac.kr

**Abstract:** *Sophora koreensis* is an endemic species of Gangwon-do, Korea, that has a variety of applications for foods and for folk remedies. Here this research analyzed and compared compounds present in leaves, stems, and roots of *S. koreensis* collected from three different habitats in Chuncheon, Inje and Yanggu in South Korea. This research also analyzed soil microorganisms present in the three habitats to determine the correlation between the compound and microorganisms. *N*-methylcytisine was the most common compound in all three habitats, but the amounts varied with Chuncheon having the highest amount (509 mg/L), followed by Yanggu and Inje(102 mg/L and 39 mg/L, respectively). The composition of microorganisms also varied by habitat. Yanggu, Inje, and Chuncheon had 1013, 973, and 814 taxa, respectively. According to the phylogenetic relations, the composition of the soil microorganisms in Chuncheon was significantly different from the other two. It contained more PAC000121_g (Solibacteres), major taxa in all three habitats (14% in Chuncheon). In contrast less Opitutus minor taxa was found than Yannggu and Inje. The correlation between the soil microorganism *N*-methylcytisine was analyzed. Among these microorganisms, Paraburkholderia had a positive correlation with *N*-methylcytisine. Meanwhile, Rhizomicrobium, CP011215_f (Paceibacter), KB906767_g (Solibacteres) and Opitutus negatively correlated with *N*-methylcytisine. The results suggested that soil microorganisms in the habitats influenced the variations of the *N*-methylcytisine.

**Keywords:** *Sophora koreensis*; compound; habitat; *N*-methylcytisine; microorganism





## 1. Introduction

*Sophora koreensis* NAKAI is a multi-stemmed shrub that grows wild at about 100~300 m above sea level in the Gangwon-do province of Korea. The plant is endemic in Korea [1] and was first collected in Bukcheong, South Hamgyoung Province, by Ishidoya and Taehyeon in 1918. In 1919, Nakai termed the plant *Sophora koreensis*, a genus of *Sophora* [2] and classified it as endemic in 1923 [3,4]. Although subsequently classified as a species belonging to the genus *Echinosophora* [3], *S. koreensis* was since reclassified back in the genus of *Sophora* [5]. The species grows well at the foot of mountains in areas that have good sun exposure and sandy loam soil. Despite its good adaptation in these regions, due to deforestation the natural habitats of *S. koreensis* are now restricted to only a few areas including Chuncheon, Yanggu, Hongcheon and Inje of Gangwon-do [6]. *S. koreensis* displays clusters of bright yellow flowers and its low-growing habits that make this plant suitable for gardening and survival in dry soil and for revegetation of sloped areas in particular.

*S. koreensis* roots have been used in oriental medicine to treat gastrointestinal diseases [7]. Antioxidant compounds in *S. koreensis* roots alleviates aging of skin exposed to common stresses such as ultraviolet rays. Lupin alkaloids such as *N*-(3-oxobutyl) cytisine, lupanin and 5,6-dehydrolupanine have been extracted from the above-ground (leaf and stem) and underground (root) portions of *S. koreensis* [8,9], whereas {−}-N-ethylcytisine, {−}-methyl 12-cytisineacetate and Ethyl 12-cytisineacetate have been extracted from its

flowers [10]. In another study, triterpenoidal alkaloid [11], tetracosanol and docosanol were isolated from ether and ethyl acetate extracts of plant tissues [12]. Saponin, an organic chemical similar to ginseng used in dietary supplements, has also been isolated [13]. These studies mainly focused on the types of extracted compounds and extraction methods. Except for the alcohololysis reaction of four extracted flavonoids [14] and their usefulness as livestock feed [15], few studies has examined the functional activity of *S. koreensis* compounds.

Natural products, especially those derived from plants, are preferred over synthetics compounds for health and beauty applications as evidenced by large increases in the market size for cosmetics and supplements containing natural compounds. As such, there is substantial interest in selecting, researching and commercially developing native plants. Multiple efforts are underway to find active substances from native plants for food and medicine and to develop new products. [16–18]. With the adoption of the Nagoya Protocol, bio-industries are seeking indigenous plants for use as raw materials. In this context, *S. koreensis* is a promising plant species for natural products.

The growth environment of plant species can greatly affect the production of secondary metabolites, which can differ by habitats due to abiotic factors such as saline soil [19–22]. In addition, soil microorganisms may affect plant growth and secondary metabolite synthesis. In agriculture, research on the association of microorganisms with growth promotion and compound amplification is being actively conducted to increase yield and industrialization [23]. Findings from these studies can provide the basis for establishing good cultivation that will contribute to stable production of bioactive compounds by plants. For *S. koreensis* in particular, the leaves and stems represent most of the biomass, but to date only the roots have been for different applications. This focus on the roots has important implications for *S. koreensis* since this species reproduces asexually from roots rather than producing seeds, even though it does flowers. Thus, consumption of the roots may accelerate *S. koreensis* extinction.

For this reason, this research analyzed the most abundant compounds in leaves, stems and roots from *S. koreensis* grown in three different habitats. This research also investigated the soil microorganism community of each habitat as a biotic factor that could influence the biosynthesis of organic compounds.

## 2. Materials and Methods

### 2.1. Plant Collection and Soil Sampling

Plants materials (3 samples per area) were collected from three habitats: Chuncheon (N 37°54′19.36′′ E 127°46′21.44′′), Yanggu (N 38°04′15.0′′ E 128°02′26.8′′) and Inje (N 38°02′15.9′′ E 128°09′32.6′′), Gangwon province. Whole *S. koreensis* plants were harvested in July. Concurrently, soil samples were taken near the roots of the harvested plants. Soil cores (diameter 2 cm), 15–30 cm deep, were collected from three randomly selected locations within each plot on site. Soil samples from three separated areas in each habitat were collected and deposited in a container that was stored at a cool temperature. Plants were divided into leaves, stem and roots that were used for the analysis. In order to compare the soil of habitat, greenhouse soil used for transplantation (nursery (peat and perlite mixture):sand = 7:3) was also analyzed as control (3 samples).

### 2.2. Analysis of Compounds in Leaves, Stems and Roots

The leaves, stems and roots were air-dried in a drying oven at 37 °C. Milled powder (100 mg) from the samples was soaked in 100% methanol (0.5 mL) and sonicated for 30 min at a constant frequency of 20 kHz at 40 °C. The samples were centrifuged ($15,000\times g$ for 10 min) and the supernatants were filtered using a SepPak C- 18 cartridge (Waters Co., Milford, MA, USA). Quantitative analysis of the main peak (*N*-methylcytisine) from *S. koreensis* samples was performed using gas chromatography-mass spectrometry (GC-MS; Agilent 5975C and 7890A, Wilmington, DE, USA). *S. koreensis* leaf and shoot extracts were also analyzed by GC-MS. For GC-MS analysis, an aliquot (5 μL) with a split injection (5:1) was analyzed by a gas chromatograph (Agilent 7890A, Wilmington, DE, USA) linked to

an inert MSD system (Agilent 5975C, Wilmington, DE, USA), with a Triple-Axis detector and equipped with an HP-5MS capillary column (30 m × 0.25 mm, 0.25 mm film thickness). The injection temperature was 250 °C. The column temperature program was: 150 °C for 5 min, increase to 300 °C at a rate of 5 °C min$^{-1}$ and hold at 300 °C for 20 min. The carrier gas was He and the flow rate was 1.2 mL min$^{-1}$. The interface temperature was 300 °C with a split/splitless injection (10:1). The temperature of the ionization chamber was 250 °C and the ionization mode was electron impact at 70 eV. Standard compounds for *N*-methylcytisine were purchased from Sigma-Aldrich Co. (Saint Louis, MO, USA). [24].

### 2.3. Soil Microorganism Community Analysis

Total DNA was extracted from 100 mg soil sample using a I-genomic soil DNA extraction Mini kit (iNtRON Biotechnology, Inc., Seongnam-Si, Korea). Primers were prepared to compare the bacterial community in the samples. The V3–V4 region was amplified using the primers in the Table 1. An Illumina adapter and locus-specific sequence was added to primers. Up to 3 NBases were added to primer since the region has many variations among bacteria [25].

**Table 1.** 16S V3–V4 region amplification primer.

| Primer | Sequence (5 > 3) |
| --- | --- |
| 16S v34_F | TCGTCGGCAGCGTCAGATGTGTATAAGAGACAGCCTACGGGNGGCWGCAG <br> TCGTCGGCAGCGTCAGATGTGTATAAGAGACAG**N\***CCTACGGGNGGCWGCAG <br> TCGTCGGCAGCGTCAGATGTGTATAAGAGACAG**NN**CCTACGGGNGGCWGCAG <br> TCGTCGGCAGCGTCAGATGTGTATAAGAGACAG**NNN**CCTACGGGNGGCWGCAG |
| 16S v34_R | GTCTCGTGGGCTCGGAGATGTGTATAAGAGACAGGACTACHVGGGTATCTAATCC <br> GTCTCGTGGGCTCGGAGATGTGTATAAGAGACAG**N**GACTACHVGGGTATCTAATCC <br> GTCTCGTGGGCTCGGAGATGTGTATAAGAGACAG**NN**GACTACHVGGGTATCTAATCC <br> GTCTCGTGGGCTCGGAGATGTGTATAAGAGACAG**NNN**GACTACHVGGGTATCTAATCC |

\* N; NBase.

The following Polymerase Chain Reaction (PCR) (GeneAmpR PCR System 9700) conditions were used: initial denaturation at 95 °C for 30 s, followed by 25 cycles of primer annealing (55 °C for 30 s) and extension (72 °C for 30 s) and a final elongation at 72 °C for 5 min. PCR amplification products were electrophoresed on 1% agarose gels (Mupid-21; COSMO Bio Co., Tokyo, Japan) and analyzed with a Bio-Rad Gel Documentation System(Bio-Rad, Berkery, CA, USA).

16S V3–V4 PCR purification was performed under the following methods: Dispense 0.02 mL of AMpure(Beckman Coulter, Pasadena, CA, USA) beads into each well, mix by pipetting and leave at room temperature for 5 min. Collect beads by centrifugation for 30 s. Put the plate on a magnetic stand, wait for about 3 to 5 min until the supernatant becomes transparent and then remove the PCR supernatant. Add 0.2 mL of 80% ethanol and remove 80% ethanol after 30 s. Repeat once more. Spin down for 30 s to remove 80% ethanol completely and dry for 5 to 10 min. Put 0.0515 mL D.W, leave it for 5 min, put it on a magnetic stand and leave it for about 3–5 min until the supernatant becomes transparent. Transfer 0.05 mL of the PCR supernatant to a new plate. After measuring the concentration of Tacan 800 pro, normalize (20 ng/uL). After 16S V3–V4 PCR purification, Index PCR was performed. Index PCR was performed under the following conditions: initial denaturation at 95 °C for 30 s followed by 8 cycles of primer annealing at 55 °C for 30 s and extension at 72 °C for 30 s with a final elongation at 72 °C for 5 min. PCR amplification products were electrophoresed as described above. Index PCR purification was also carried out as described above using the 16S V3–V4 PCR purification method. The size of amplicon library was about 564 bp.

Sequencing of the bacterial communities in 12 soil samples was performed using Miseq ver 3 reagent and 2 × 300 bp with the 600-cycle kit (Illumina, San Diego, CA, USA). Bacterial community analysis using 16 S nucleic sequence was analyzed by EzBioCloud

pipeline software(Chunlab, Inc., Seoul, Korea) [26]. Taxonomy database, 16 S database version PKSSU4.0 (https://www.Ezbiocloud.net/, accessed on 20 May 2022) of EzBioCloud was used. Sequences of bacterial community were classified by quality and low quality (<Q value 20, non-target, chimeric) reads were excluded before analysis of Operational Taxonomic Units (OTUs) based on 97% homology among the good reads. For the species diversity index, abundance-based coverage estimator (ACE) and Chao statistical programs were used to describe species abundance statistics. In addition, the Shannon statistical program was used to statistically measure diversity indices by species uniformity.

The rarefaction curves generated by QIIME (v. 1.9.1) were used to evaluate the depth of sequence, species richness and species uniformity [27]. For UPGMA (Unweighted Pair-Cluster Method using Arithmetic Averages) analysis to investigate the similarity of each area, dendrograms were generated between each sample by NTSYS-pc software(v. 2.02 K, Applied Biostatistics, Inc., New York, NY, USA) [28].

### 2.4. Statistical Analysis

Statistical analysis was performed with R software (R version 4.0.3, GNU General Public License, New Zealand). ANOVA and Tuckey's HSD test post hoc test ($p < 0.05$) were used to compare microbial diversity. Pearson's correlation coefficient (r) and significance ($p < 0.05$) were confirmed between the concentration of the secondary compound from *S. koreensis* by habitat and major communities of different microorganism genus in the soil.

## 3. Results and Discussion

### 3.1. Comparison of Secondary Compounds in Parts of S. koreensis Plants from Different Habitats

GC-MS analysis was performed with extracts of *S. koreensis* plant from different habitats. For the above-ground parts of the plant, the largest peak was seen at retention time of 15 min (Figure 1a) that corresponding to the retention time and mass fraction of and *N*-methylcytisine (Figure 1b,c). This result is consistent with a previous study by Murakoshi et al. (1982) [9] that optimized a gas chromatography system to analyze Fabaceae extracts for the content of *N*-methylcytisine, which was shown to be the secondary compound in flowers of this plant type. Furthermore, the results are in agreement with earlier [8,9] showing that lupine alkaloids are the main substance in Fabaceae. *N*-methylcytisine is a type of lupine alkaloid. The best-known lupine alkaloid is sparteine, which is bitter and acts to blocking sodium channels that in turn causes vasoconstriction, cardiac hyperactivity, pulse control and increased blood pressure. Sparteine is pharmacologically similar to nicotine and coniin [29] although its effects are weaker. Plant extracts and some alkaloids were shown to exhibit cytotoxic activity against cancer cell lines [30]. Meanwhile, treatment with high concentrations of *N*-methylcytisine in animal models induced increased blood pressure, small intestine irritation and convulsions [31] as well as hyperglycemia. Thus, careful modulation of *N*-methylcytisine is needed to balance its beneficial and negative pharmacological effect.

In preliminary analysis, each part of the plant had different *N*-methylcytisine concentrations, with 10- and 100-fold higher concentrations seen in leaves relative to the stem and root, respectively (values shown are in terms of peak volume). Due to these high concentrations, above-ground parts of *S. koreensis* are more likely not to be consumed. The *N*-methylcytisine concentration differed by habitats. Leaves from Chuncheon had the highest concentration of *N*-methylcytisine, 509 mg/kg, which was substantially higher than that seen in samples from Yanggu (102 mg/kg) and Inje (39 mg/kg) (Figure 2). Previous studies have investigated the effects of various environmental factors, such as temperature, precipitation, and altitude on compounds present in plants. For *Paeonia lactiflora* Pall, changes in albiflorin and paeoniflorin content as well as root dry weight from soil available $P_2O_5$ were confirmed using UPLC. Another study showed differences among plants depending on local soil characteristics [32]. For *S. koreensis*, the soil type and forest composition of natural habitats are reported to be highly similar [6]. The species propagates vegetatively and thus the genetic variability among the habitats would be expected to be

very low [33]. As such, variation in the contents of the same compound may be influenced by other factors than those associated with the environment.

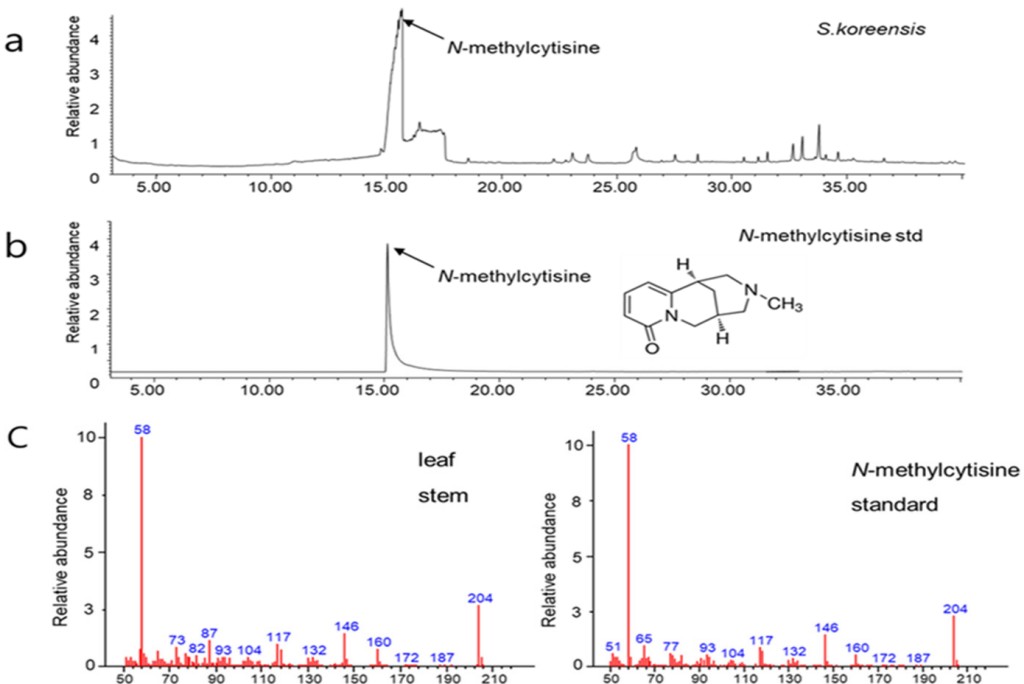

**Figure 1.** GC-MS analysis of *S. koreensis* in Chuncheon and *N*-methylcytisine standard (**a**) GC chromatogram of extract from *S. koreensis* leaf and stem. (**b**) GC chromatogram of the authentic standard of *N*-methylcitisine. (**c**) Mass fraction of *N*-methylcytisine peak in *S. koreensis* and *N*-methylcytisine standard.

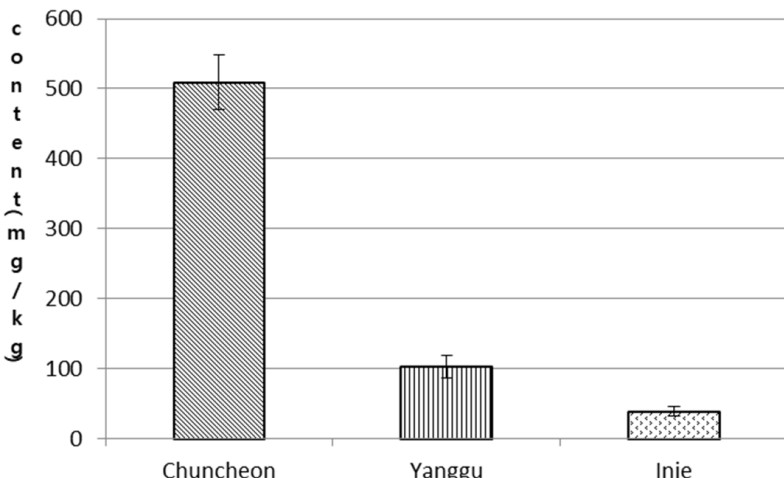

**Figure 2.** *N*-methylcytisine concentration of *S. koreensis*.

### 3.2. Soil Microbial Analysis by Habitat

Soil microorganisms are one factor that affects synthesis of secondary metabolites in plants. As the *N*-methylcytisine content of *S. koreensis* significantly differed by habitat, this study analyzed its relationship to microbial communities in soil samples from each habitat.

The alpha diversity of the soil sample community analysis for the 12 samples was determined (Table 2). The OTU and diversity index were high in greenhouse soil samples that are commonly used for transplantation. In terms of differences in OTUs among soil samples, the greenhouse transplantation soil OTU was 1.47 times higher than that for the Chuncheon, Inje and Yanggu soil samples. On the other hand, the average OTUs

for soils from the three regions was 3061.22, which was lower than that of the potting mix soil. Yanggu 2 had the highest OTU (3742) and Chuncheon 1 had the lowest (2167). Yanggu 2 also had the highest phylogenetic diversity (3786), while Chuncheon 1 had the lowest (2554). The abundance of soil bacteria varied by habitat. ACE was highest for the greenhouse soil sample at 4980.33. Inje and Yanggu had similarly high values (4011.67 and 4009.72, respectively), whereas the ACE for Chuncheon was 3095.61, which was relatively low compared to the other two habitats. Values from Chao statistical program was also consistent with this result. The three soils of each habitat yielded comparable numbers of sequencing reads and OTUs.

**Table 2.** Soil samples Alpha diversity.

| Area | Target Reads | OTUs | ACE | CHAO | Shannon | Phylogenetic Diversity |
|---|---|---|---|---|---|---|
| C1 | 60,046 | 2167 | 2496.92 | 2375.37 | 5.56 | 2554 |
| C2 | 42,686 | 2626 | 3095.61 | 2936.56 | 6.05 | 3099 |
| C3 | 30,755 | 2454 | 2862.87 | 2755.94 | 6.35 | 2834 |
| | 44,495.67 ± 8503.86 [a*] | 2415.67 ± 133.88 [a] | 2818.47 ± 174.25 [a] | 2689.29 ± 165.39 [a] | 5.99 ± 0.23 [a] | 2829.00 ± 157.35 [a] |
| G1 | 85,014 | 4727 | 4980.30 | 4851.04 | 6.76 | 4818 |
| G2 | 92,879 | 4590 | 4802.33 | 4695.96 | 6.64 | 4635 |
| G3 | 93,790 | 4254 | 4353.61 | 4303.22 | 6.59 | 4337 |
| | 90,561.00 ± 2785.94 [b] | 4523.67 ± 140.51 [c] | 4712.08 ± 186.45 [c] | 4616.74 ± 163.03 [c] | 6.66 ± 0.05 [b] | 4596.67 ± 140.17 [b] |
| I1 | 49,775 | 3473 | 3819.37 | 3682.19 | 6.53 | 3559 |
| I2 | 34,407 | 2522 | 2831.94 | 2707.12 | 6.35 | 2769 |
| I3 | 66,734 | 3716 | 4011.67 | 3857.06 | 6.45 | 3643 |
| | 50,305.33 ± 9335.77 [a] | 3237.00 ± 364.32 [ab] | 3554.33 ± 365.43 [ab] | 3415.46 ± 357.75 [ab] | 6.44 ± 0.05 [ab] | 3323.67 ± 279.39 [a] |
| Y1 | 74,273 | 3658 | 3961.13 | 3821.48 | 6.19 | 3445 |
| Y2 | 64,172 | 3742 | 4009.72 | 3877.06 | 6.30 | 3786 |
| Y3 | 56,853 | 3193 | 3552.73 | 3404.81 | 6.08 | 3250 |
| | 65,099.33 ± 5050.05 [ab] | 3531.00 ± 170.73 [b] | 3841.19 ± 144.91 [bc] | 3701.12 ± 149.02 [b] | 6.19 ± 0.06 [ab] | 3693.67 ± 156.63 [a] |

* The different letters in column indicate significantly difference at $p < 0.05$ by Tuckey HSD test. C, Chuncheon; G, Greenhouse, I, Inje; Y, Yanggu.

The rarefaction curve indicates the correlation between the number of nucleic sequences and the number of OTUs. In Figure 3, the soil of greenhouse is higher microbial diversity than that of the habitats of *S. koreensis*. The soil of Chuncheon shows lowest in all samples. It can be seen that Yanggu and Inje have a similar level of diversity.

UPGMA clustering analysis also showed that the bacterial community in the potting mix soil differed markedly from the soil community in Chuncheon, Yanggu and Inje samples. In one of the three soil samples from Chuncheon, a community pattern similar to that for Inje was observed, whereas Inje and Yanggu soils had similar cluster patterns (Figure 4) From this result, it can be inferred that the microbial structure of the soil is different since *S. koreensis* grows wild. Still, the difference between the habitats is due to the small number of samples analyzed. Hence, it is necessary to analyze and compare the structure of the microbial community in more soil samples in the future.

*3.3. Comparison of Bacterial Communities at the Genus Level*

Based on the nucleic sequence, comparative analysis of microorganisms was performed at the genus level. The group in which the sum of the ratios of all samples accounted for less than 4.5 was classified as etc. For bacterial community analysis at the genus level, 1845 taxa were investigated and 25 representative taxa were diagrammed (Figure 5). The bacterial community composition in the sample can be listed in the order of genus content as follows: PAC000121_g (Solibacteres) (14.53%~0.485%), Rhizomicrobium (4.24%~2.3%), PAC001932_g (Spartobacteria) (5.15%~0.18%), Sphingomonas (8.01%~0.73%), Solibacter (3.58%~0.84%), KB906767_g (Solibacteres) (3.43%~1.33%), PAC000030_g (Solibacteres) (6.87%~0.05%), PAC002252_g (Spartobacteria) (3.79%~0.01%), Bradyrhizobium (3.12%~1.11%), Tepidisphaera (3.51%~0.52%), which were investigated in that order. The genus-level communities that appeared in each soil were greenhouse (1467 taxa), Yanggu (1013 taxa), Inje (973 taxa) and Chuncheon (814 taxa).

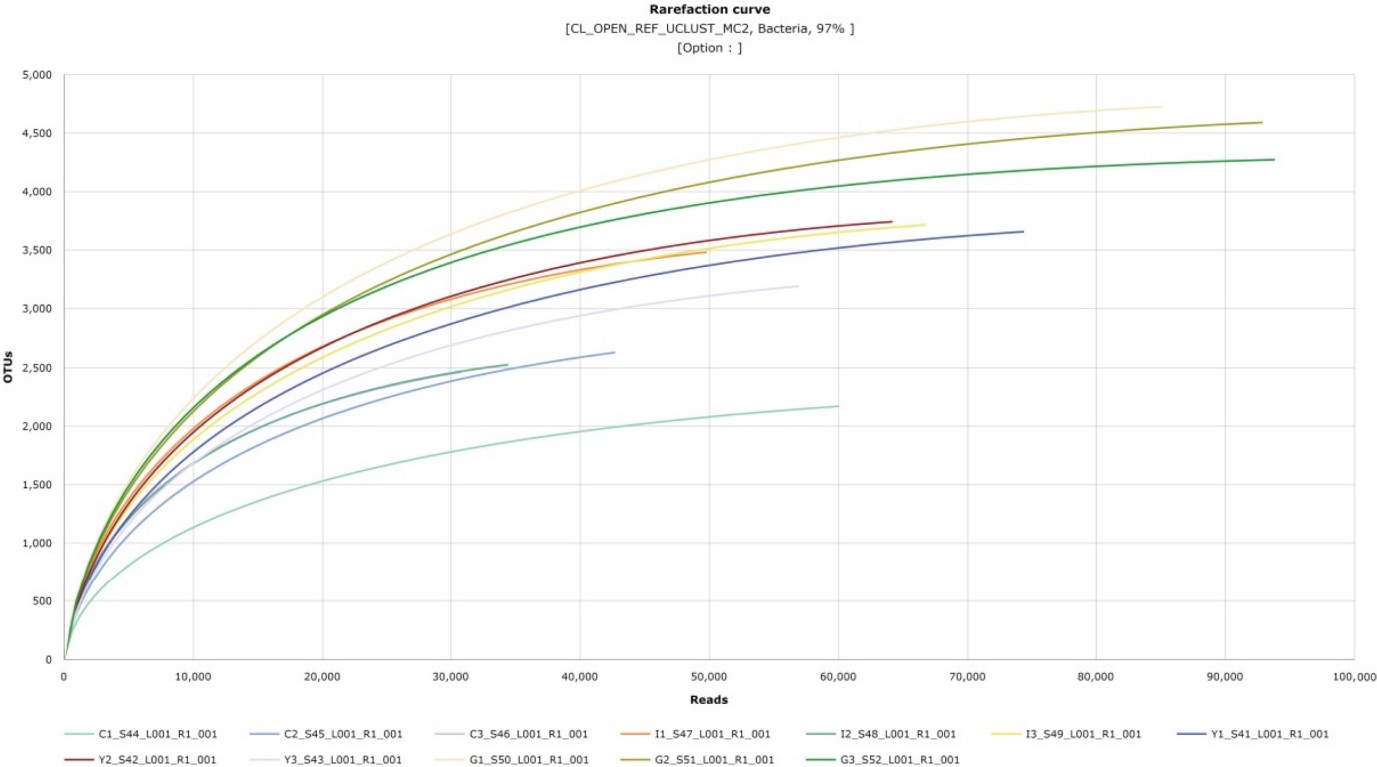

**Figure 3.** Rarefaction curves of OTUs at different areas.

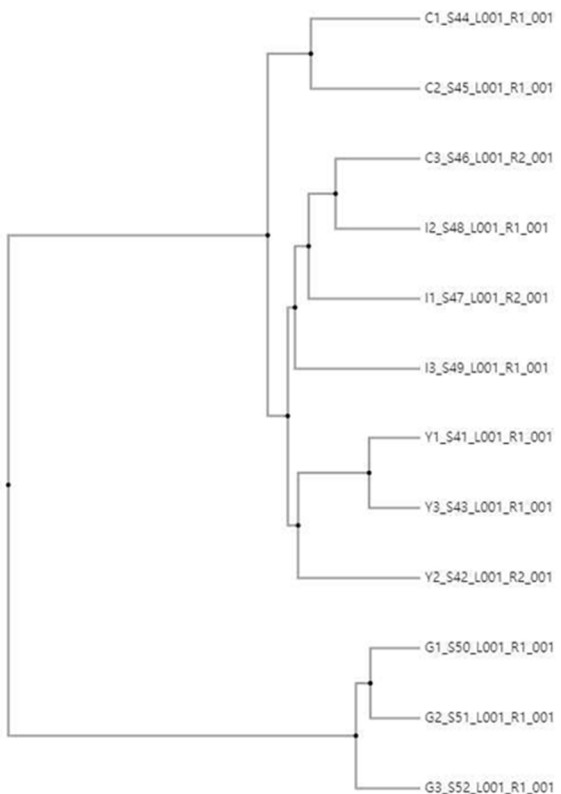

**Figure 4.** UPGMA analysis among soil samples from three different habitats and greenhouse (C, Chuncheon; G, Greenhouse; I, Inje; Y, Yanggu).

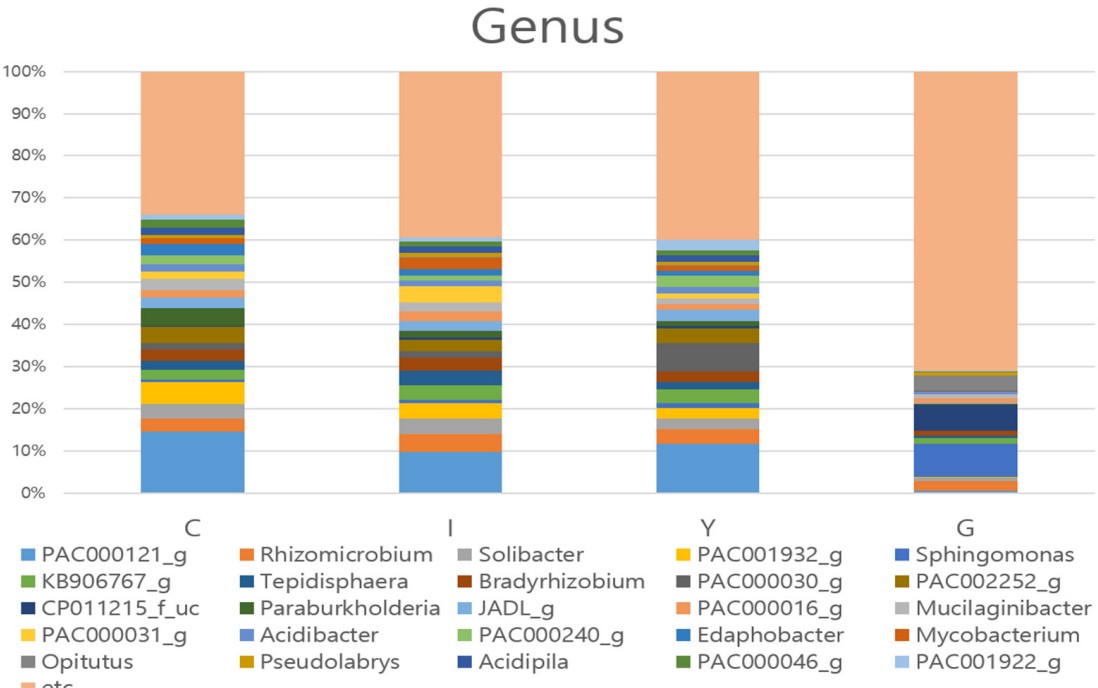

**Figure 5.** Relative abundance of cluster analysis by genus level soil (C, Chuncheon; G, Greenhouse, I, Inje; Y, Yanggu).

Except for the microbiome of the potting mix soil, the frequency of PAC000121_g (Solibacteres), Rhizomicrobium, Solibacter and PAC001932_g (Spartobacteria) was higher than any other taxa across all habitats tested. It contained more PAC000121_g (Solibacteres), major taxa in all three habitats (14% in Chuncheon). Soil from Chuncheon had abundant PAC001932_g (Spartobacteria), whereas Inje and Yanggu had high amounts of KB906767_g (Solibacteres). Unlike other regions, soil from Yanggu contained large amounts of PAC000030_g (Solibacteres) and Inje soil had a high percentage of PAC000031_g (Acidobacteriia). The potting mix soil contained bacterial communities that were substantially different from the soil in the habitats.

### 3.4. Correlation among Soil Microorganisms and Secondary Compounds

Bacteria and fungi have functional importance for plants. Plant-associated microorganisms have an interactive relationship in which specific microorganisms can stimulate biosynthesis and signaling pathways in the host plant to produce specific metabolic compounds [34,35]. Previous studies have shown that plant root-associated microorganisms stimulate root leaching of systematically induced metabolites that can affect levels of root transcripts and metabolites [36]. A study conducted in Paraná, Brazil, confirmed that the types and levels of concentrations of compounds produced by five study species were the same in the soils of the seasonal semideciduous forest and the lowland ombrophilus dense forest, although the levels of these compounds differed between the sites. The type of tannin of *Schinus terebinthifolius* also differed between these forests [37].

In microbial community present in the soil where *S. koreensis* grows wild, genus-level bacterial communities were selected for comparison with *N*-methylcytisine using Pearcon's correlation analysis. A negative correlation (values in parentheses) with *N*-methylcytisine was seen for the following bacteria in descending order: KB906767_g (Solibacteres) (−0.82), Rhizomicrobium (−0.75), CP011215_f (Paceibacter) (−0.72), Opitutus (−0.69). Edaphobacter and Acidipila was positively correlated with a correlation index of 0.80. Edaphobacter also was positively correlated with PAC000046_g (Solibacteres) (0.66; Figure 6). *N*-methylcytisine concentration was negatively correlated with Rhizomicrobium and CP011215_f (Paceibacter) with correlation coefficients of −0.75 and −0.72, respectively and between the two there

was a positive correlation of 0.90 (Figure 7). KB906767_g (Solibacteres) negatively correlated with the concentration of *N*-methylcytisine (−0.82; Figure 8). KB906767_g (Solibacteres) and PAC000030_g (Solibacteres) had a positive correlation of 0.76 and solibacter had a positive correlation with PAC001932_g (Spartobacteria). JADL_g (Alphaproteobacteria) was positively correlated with Bradyrhizobium and PAC002252_g (Spartobacteria) with correlation coefficients of 0.84 and 0.88, respectively (Figure 9). *N*-methylcytisine concentration was negatively correlated with opitutus (−0.69). There was positive correlation among the three genera, Bradyrhizobium, JADL g and Tepidisphaera. The Bradyrhizobium genus is the most common microorganism in soil worldwide [38] and plays a critical role in nitrogen fixation and soil fertility [39]. Many studies have demonstrated that Bradyrhizobium can be used in toxicity studies [40–43]. Opitutus negatively correlated with *N*-methylcytisine (−0.69; Figure 10). Mucilaginibacter had positive correlation with PAC000240_g (Gammaproteobacteria) (0.73; Figure 11). In contrast, Paracurkholderia had negative correlation with *N*-methylcytisine (0.64). The genus Opitutus belongs to the phylum Verrucomicrobia, which is involved in the nitrogen cycle in which bacteria promote reduction of nitrate to nitrite [44]. Previous studies showed that steppe diversity is determined by the relative abundance of the predominant Verrucomicrobia phylum. This study also showed positive correlations with various genes related to carbohydrate metabolism, but negative correlation with genes associated with nitrogen metabolism and cell division, which may be due to their role in the carbon cycle [45].

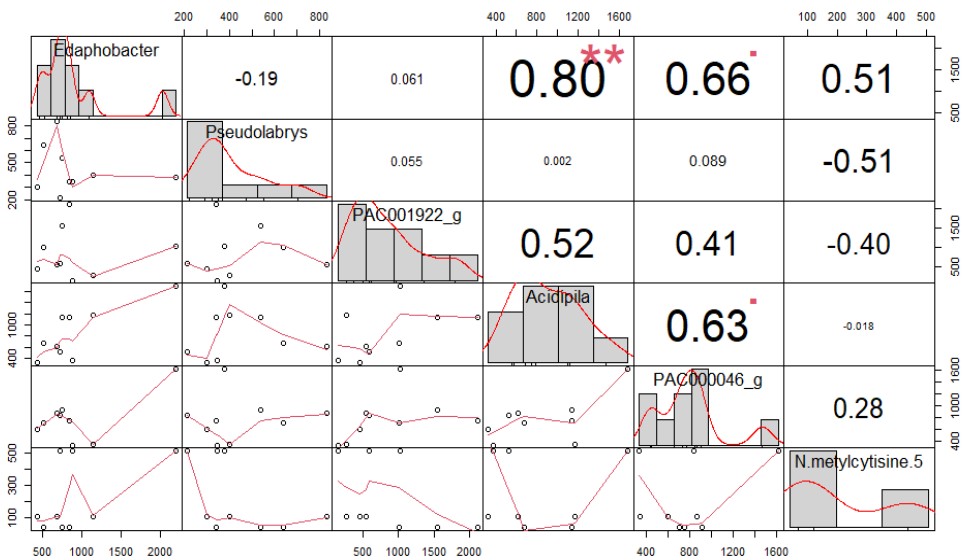

**Figure 6.** Pearson's correlation coefficient among soil microorganisms (Edaphobacter, Pseudolabrys, PAC001922, Acidipila, PAC000046) and secondary compound of *S. koreensis* * Significance between the parameter are indicated * $p < 0.05$, ** $p < 0.01$, indicates significant correlation ($p < 0.05$).

There were differences in the number and the type of microorganisms among the habitats and some microorganisms are significantly correlated with the concentration of the secondary compound, *N*-methylcytisine. *S. koreensis* generally grows in acidic (pH 4.2–5.9) forest soils [46]. Phylum acidobacteria (PAC000121_g, Edaphobacter, Acidipila, PAC000046_g, Solibacter, PAC000031_g, PAC000030_g, KB906767_g, Mycobacterium) is high in the soil in the habitats. Phylum acidobacteria are reported to be abundant in acidic soils [47–49]. The comparison with the soil in greenhouse as a control in this research showed that soil microorganisms are totally different from the acidic soil from habitats. Also, biotic environment of plants, especially soil microorganisms affect plant growth promotion or secondary metabolites production [50]. The growth of *S. koreensis* was also different for each habitat (in preliminary investigation, not shown). However further research is needed since the microorganisms of *S. koreensis* can affect the growth and secondary metabolites other than *N*-methylcytisine.

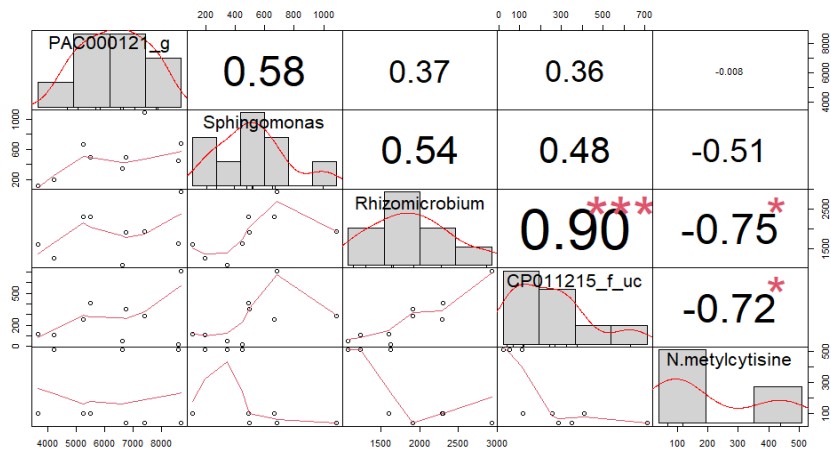

**Figure 7.** Pearson's correlation coefficient among soil microorganisms (PAC000121, Sphingomonas, Rhizomicrobium, CP011215) and secondary compound of *S. koreensis* * Significance between the parameter are indicated * $p < 0.05$, *** $p < 0.001$, indicates significant correlation ($p < 0.05$).

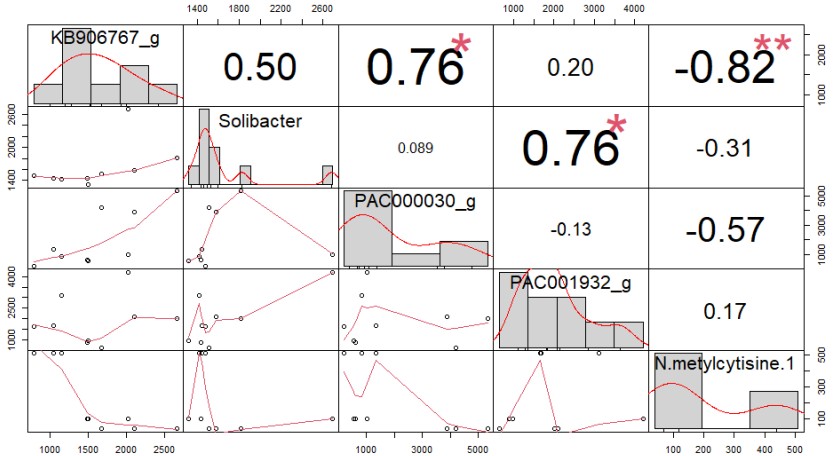

**Figure 8.** Pearson's correlation coefficient among soil microorganisms (KB906767, Solibacter, PAC000030, PAC001932) and secondary compound of *S. koreensis* * Significance between the parameter are indicated * $p < 0.05$, ** $p < 0.01$, indicates significant correlation ($p < 0.05$).

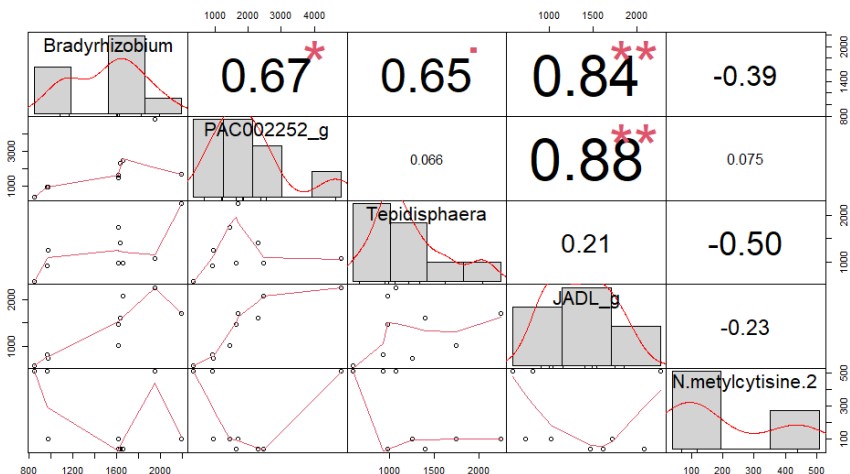

**Figure 9.** Pearson's correlation coefficient among soil microorganisms (Bradyrhizobium, PAC002252, Tepidisphaera, JADL) and secondary compound of *S. koreensis* * Significance between the parameter are indicated * $p < 0.05$, ** $p < 0.01$, indicates significant correlation ($p < 0.05$).

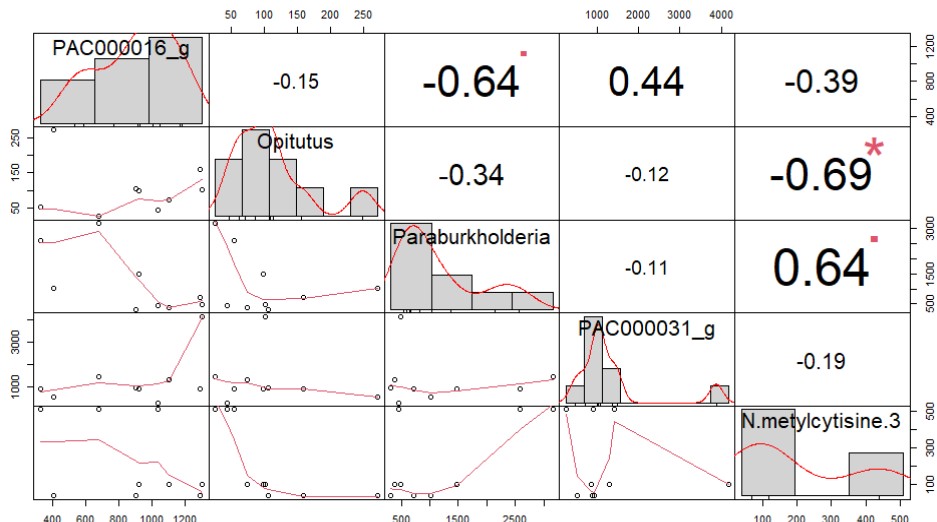

**Figure 10.** Pearson's correlation coefficient among soil microorganisms (PAC000016, Opitutus, Paracurkholderia, PAC000031) and secondary compound of *S. koreensis* * Significance between the parameter are indicated * $p < 0.05$, indicates significant correlation ($p < 0.05$).

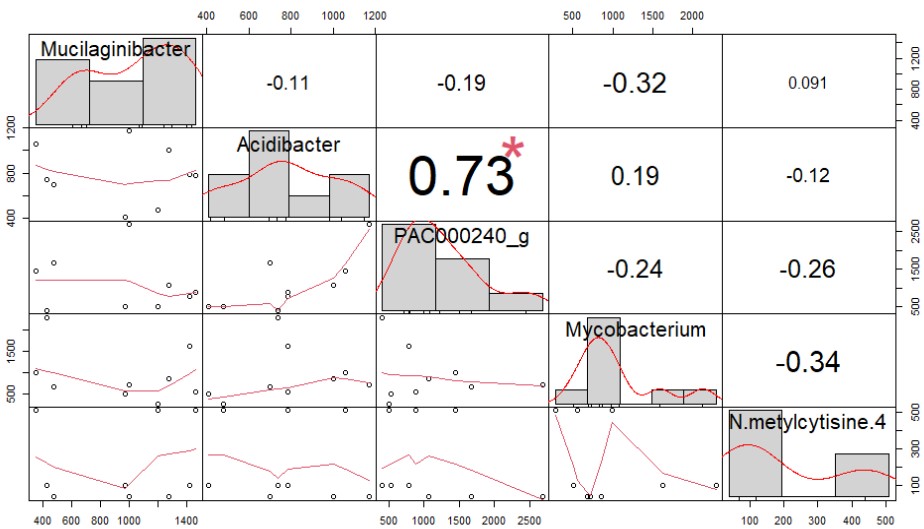

**Figure 11.** Pearson's correlation coefficient among soil microorganisms (Muciaginibacter, Acidibacter, PAC000240, Mycobacterium) and secondary compound of *S. koreensis* * Significance between the parameter are indicated * $p < 0.05$, indicates significant correlation ($p < 0.05$).

In conclusion, *N*-methylcytisine was the most abundant compound in *S. koreensis* and the concentrations varied by habitat. Analysis of soil microorganisms confirmed that there might be differences depending on the area, even in the same plant community. It suggested that composition of microbial community may be the cause of the difference in the concentrations of secondary metabolites. The *N*-methylcytisine content of *S. koreensis* had a high negative correlation with Rhizomicrobium, CP011215_f (Paceibacter), KB906767_g (Solibacteres) and Opitutus in soil and a high positive correlation with Paraburkholderia. Future studies should examine the effect of inoculation with these organisms on the content of *N*-methylcytisine.

**Author Contributions:** Investigation, H.L.; writing—original draft preparation, H.L.; writing—review and editing, E.J.C. All authors have read and agreed to the published version of the manuscript.

**Funding:** This work was supported by Korea Forestry Service (Korea Forestry Promotion Institute) under R&D Program Forest Science Technology (Project No. "2021320A00-2122-AA03"). This

work was supported "Forest Bioresources(*Sophora koreensis* Nakai) collection, conservation and characteristic assessment (202202210001)" by National Forest Seed and Variety Center.

**Institutional Review Board Statement:** Not applicable.

**Informed Consent Statement:** Not applicable.

**Data Availability Statement:** Not applicable.

**Conflicts of Interest:** The authors declare no conflict of interest.

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
