# Peer review of "Correlation between the Concentration of Secondary Metabolites and Soil Microorganisms in Sophora Koreensis Nakai from Different Habitat"

_forests, doi:10.3390/f13071079_

Round 1
Reviewer 1 Report
The manuscript by Lee et al. seems to represent an explorative research about possible correlations between the production of secondary bioactive compounds by an endemic plant species, and bacterial community structure in soil. However, the findings are not properly discussed: positive and negative correlations are described, but their meaning remains completely unclear, if any. Moreover, soil bacterial communities were analysed, which are less tightly correlated to plant functions than rhizospheric or phyllospheric communities. The reason of this choice is not addressed at all. Microbiological molecular methods are not clearly described (see below for details). For these reasons, I don’t feel that this manuscript would be suitable for publication in the present form.
Detailed comments:
-Title: the meaning of “main compound” is not easy to understand unless you read also the abstract. I think it would be more effective to use more precise words, such as “secondary metabolites” or “bioactive compounds”. Please check the spell of Sophora koreensis.
-Paragraph 2.1: how many replicate samples of plants were collected at each site? Was the collected soil bulk soil or rhizospheric soil?
-Line 103: why was soil freeze-dried rather than simply frozen?
-Lines 105-109: what the authors mean with “mutations in 3 or 4 sections of bacterial 16s ribosomal DNA”? Which hypervariable regions were the target? Reference to used primers is missing: which primers are they? Please use the usual names according to E. coli numeration. How long is the amplicon? Why were Ns inserted between the adaptor and the primer?? This protocol is very unusual so this choice should be explained and supported.
-Line 112: what is a mutation section?
-Lines 121-122: PCR product purification is not described at all.
-Line 124: I guess that “600 cycle mode” means “2 x 300 bp paired-end protocol”. Is that right?
-Which database was used for taxonomic classification of OTUs? How? This part of Methods is completely lacking.
-Line 135: was correlation analysis performed on OTUs or on genera? It is not clear either from the Methods or from the Results.
-Is Figure 1a related to leaves, stems, or roots? From which site? Are the other results comparable to this one?
-Line 157: why are results expressed in terms of peak volume, and not related to dry weight of plant material?
-Line 160-161: why are these concentrations expressed in mg/L? Liters of what? Why are they not referred to dry weight? How many replicates were analyzed for each site? I cannot see either standard deviations in the text, or error bars in Figure 2.
-Paragraph 3.2: why is this sentence not a part of paragraph 3.3?
-Lines 183-191: how was the number of OTUs determined? If authors want to compare this result among samples, this should be achieved either through a rarefaction analysis, or by computing OTUs on a subset of sequences including the same number of reads for all samples.
-Line 200: this is not a cluster analysis
-Lines 202-203: does this mean that 1845 genera were present in the overall dataset? How were 50 of them chosen for graphical representation? Figure 3 is hardly readable. I would suggest representing a lower number of genera and grouping the others in a single entry.
-Lines 202-217: when genera are identified by a code, please refer to a known taxon for better clarity. What is “genus appearance”? It does not seem to refer to abundance here. What do the numbers in parentheses mean?
-Paragraph 3.6: it appears that correlation analysis was performed on the abundance of selected genera: how were they chosen? Figures 5-10 are not so clear to read and understand; why not using a simpler table reporting correlation coefficients and their statistical significance? Moreover, the meaning of these correlations is not actually discussed. Rhizospheric bacteria are cited multiple times, but it is not clear if analyzed communities in this manuscript were from bulk soil or rhizospheric soil. Overall, discussion should be significantly improved.
Reviewer 2 Report
Dear authors, I have read with interest the manuscript entitled "Correlation between the concentration of main compound and soil microorganisms in Sophora korrensis Nakai from different habitat". Your work is interesting and some changes are required to improve it.
General comments - please remove all the personal expressions as "we" "our" etc. Make the text more impersonal. Try to avoid long sentences, where is the case, split the sentence in two or multiple sentences based on the number of idea presented. Pay attention to italic form of genus and species in all the text (e.g. line 31)
Line 32 - The species grows well at the foot of mountains in areas that areas that have good sun exposure and sandy loam soil
Line 71-74 - This part is very important. Make it as a separate paragraph, with a clear aim of the study and all the objectives you had in research. You need to present each objective because it will prepare the reader for the Results and Discussion sections.
Mat and Meth - present the number of samples from each location: both for soil and plant. Present the coordinates of habitats, and their differences - altitude, soil type etc. Add the packages for data analysis.
Results and Discussion
3.2. Soil Microbial Analysis by habitat - this sub-section should be expanded or removed
3.3. Miseq analysis - you present here a cluster analysis. Insert in Data analysis a part to describe the package used for it. Also, for each parameter of Alpha diversity indicate the package and paper used as a methodology.
Present the averages and a data analysis - ANOVA and multiple comparison if consider useful.
3.5. Comparison of affinities among habitats - UPGMA with what software? Expand the interpretation
Rewrite the conclusion as a distinct section and provide data for the most interesting results obtained.
Round 2
Reviewer 1 Report
See attached file

Reviewer 2 Report
Dear authors, your work presents multiple improvements.
There are still some changes that need to be corrected in the text.
Rewrite the sentences between lines 10-13 and lines 71-74 in order to remove the word "we". Make these sentences more impersonal.
Author Response
Thanks for your comment. Modified according to your suggestion.